# An Integrative Pancreatic Cancer Risk Prediction Model in the UK Biobank

**DOI:** 10.3390/biomedicines11123206

**Published:** 2023-12-01

**Authors:** Te-Min Ke, Artitaya Lophatananon, Kenneth R. Muir

**Affiliations:** Division of Population Health, Health Services Research and Primary Care, School of Health Sciences, Faculty of Biology, Medicine and Health, The University of Manchester, Manchester M13 9PT, UK; te-min.ke@manchester.ac.uk (T.-M.K.); artitaya.lophatananon@manchester.ac.uk (A.L.)

**Keywords:** pancreatic cancer, polygenic score, risk prediction model, nomogram, random forest model, UK Biobank cohort

## Abstract

Pancreatic cancer (PaCa) is a lethal cancer with an increasing incidence, highlighting the need for early prevention strategies. There is a lack of a comprehensive PaCa predictive model derived from large prospective cohorts. Therefore, we have developed an integrated PaCa risk prediction model for PaCa using data from the UK Biobank, incorporating lifestyle-related, genetic-related, and medical history-related variables for application in healthcare settings. We used a machine learning-based random forest approach and a traditional multivariable logistic regression method to develop a PaCa predictive model for different purposes. Additionally, we employed dynamic nomograms to visualize the probability of PaCa risk in the prediction model. The top five influential features in the random forest model were age, PRS, pancreatitis, DM, and smoking. The significant risk variables in the logistic regression model included male gender (OR = 1.17), age (OR = 1.10), non-O blood type (OR = 1.29), higher polygenic score (PRS) (Q5 vs. Q1, OR = 2.03), smoking (OR = 1.82), alcohol consumption (OR = 1.27), pancreatitis (OR = 3.99), diabetes (DM) (OR = 2.57), and gallbladder-related disease (OR = 2.07). The area under the receiver operating curve (AUC) of the logistic regression model is 0.78. Internal validation and calibration performed well in both models. Our integrative PaCa risk prediction model with the PRS effectively stratifies individuals at future risk of PaCa, aiding targeted prevention efforts and supporting community-based cancer prevention initiatives.

## 1. Introduction

Pancreatic cancer (PaCa) is a lethal cancer with an increasing incidence and a poor survival rate. In 2020, PaCa was the 12th most common cancer and the 7th primary cause of cancer death in both sexes worldwide [1,2]. In the UK, PaCa is the 10th leading cancer, accounting for 3% of total cancer cases, with around 10,452 people diagnosed annually (years 2016–2018) [3]. Regarding cancer-related deaths, PaCa ranked as the 5th most common contributor to cancer-related mortality in the UK (years 2017–2019) [4], accounting for approximately 6% of total cancer deaths. It is challenging to diagnose PaCa at an early stage among patients with no specific symptoms, together with there being no specific screening program for PaCa, resulting in a poor survival rate and prognosis. The five-year survival rate is only approximately 7% [5]. Therefore, identifying the at-risk population in the community for early prevention is important.

In addition to the lifestyle risk factors, such as tobacco smoking or medical history-related diseases (such as diabetes mellitus), genetic predisposition is also emerging to play an important role for disease prediction [6]. Single-nucleotide polymorphisms (SNPs) are one of the common types of individual genetic variants [7], which have been used to predict the risk of developing coronary heart disease [8], diabetes [9], and cancers [10,11,12]. Various susceptible loci for pancreatic cancer have been identified from genome-wide association studies (GWAS) [13,14,15,16,17]. Currently, at least 194 variants and risk alleles associated with pancreatic cancer are listed in the GWAS Catalogue [18]. Of these, most PaCa risk loci have been found in the European population. The previous studies [19,20] have combined pancreatic cancer-associated SNPs obtained from GWAS into a polygenic risk score (PRS) to develop the risk prediction models. A case-control study [20] revealed that the highest quintile of the weighted PRS was related to increased PaCa risk (OR = 2.70, 95% CI: 1.99–3.68), compared to the lowest quintile of the weighted PRS. Another case-control study [19] showed that the top quintile of the PRS was associated with higher PaCa risk (OR = 2.25, 95% CI: 1.73–2.92) compared with the middle quintile of the PRS. Higher PRS can thus be used to identify the higher risk of PaCa-associated SNPs. Therefore, PRS could also be used for pancreatic cancer risk stratification in the population.

There are multiple models [21] that have been published associated with PaCa risk prediction. However, each model concentrated only on selected risk variables, such as symptoms-based, genetic-based, or lifestyle-related. There is still a lack of a comprehensive model built in a large prospective cohort integrating the lifestyle-related modifiable risk variables, genetic-related variables, and medical history-related variables. In this study, therefore, we aimed to establish an integrative PaCa risk prediction model using data from the UK Biobank (UKB), which could be applied in healthcare settings.

## 2. Materials and Methods

### 2.1. Study Population and Study Design

This study is a nested case-control study using the UK Biobank cohort dataset. The UKB cohort is an ongoing cohort study that recruited 502,387 participants across the UK from 2006 to 2010. The participants who withdrew from the UK Biobank study were already removed from the UKB central dataset before we obtained it [22,23]. The age at recruitment [24] for all participants ranged from 37 to 73 years old, and the last clinical outcome follow-up date of this study was 5 December 2022. The dataset contains lifestyle, genetic, and various health information, and more details on the UK Biobank can be found at: http://www.ukbiobank.ac.uk/ (accessed on 5 December 2022).

This analysis identified pancreatic cancer cases and cancer-free controls according to the criteria listed in Appendix A. Three coding systems, including the International Classification of Diseases 9 and 10 (ICD9 and ICD10) codes, and self-reported data were used. PaCa cases included all the subtypes and different anatomic parts of pancreatic malignancies to PaCa (Appendix A). Incident or prevalent cases were distinguished by the diagnosis time in relation to the study enrolment date. The total number of participants with PaCa as an incident case (diagnosis time was recorded after enrolment in all three sources) was 1402 in the UKB cohort. A participant was categorized as a prevalent case only if they were a prevalent case in at least one of three different sources and were not considered an incident case in any of the data sources (*n* = 92). Only PaCa incident cases were included in this study. Cancer-free controls were defined as the participants without any records of neoplasms, in situ neoplasms, benign neoplasms, and neoplasms of uncertain or unknown behavior (*n* = 389,027). In sum, 92 prevalent cases and 111,866 participants with records of any other neoplasms were excluded from our analysis. This study also investigated the roles of genetic component variables in the prediction model; therefore, only participants who passed sample quality control (QC) were included in this study. Therefore, there were 258,308 participants with 960 PaCa cases and 257,348 cancer-free controls included in the analyses. The flowchart of the PaCa cases and cancer-free controls selection process is illustrated in Appendix A.

The exposure variables were classified into non-modifiable, lifestyle-related modifiable, and medical history-related variables. The non-modifiable variables included gender, age, blood type, family history of bowel cancer, and polygenic score (PRS). Family history of pancreatic cancer is considered one of the risk factors for developing pancreatic cancer [25]. However, this variable was unavailable in this UKB cohort study. On the other hand, a previous study [26] advocated that a family history of bowel cancer might be associated with a higher PaCa risk, and some hereditary colorectal cancers were reported [27] to share common germline mutation genes with hereditary PaCa. Therefore, in this study, we adopted a family history of bowel cancer as a surrogate instead of a family history of PaCa [26,27] based on the evidence suggesting a link between family history of bowel cancer and pancreatic cancer. The lifestyle-related modifiable variables were tobacco smoking, alcohol intake amount, body mass index (BMI), waist-to-hip ratio (WHR), and physical activity. Medical history-related variables include pancreatitis, diabetes mellitus (DM), hepatitis B, gallbladder-related disease (cholecystitis, cholelithiasis, and cholecystectomy), *Helicobacter pylori* (*H. pylori*) infection, peritonitis, vitamin D deficiency, and systemic lupus erythematosus (SLE). The alcohol consumption classification was based on the 2018 World Cancer Research Fund/American Institute for Cancer Research (WCRF/AICR) recommendations [28,29]: men and women who consume no more than 28 g and 14 g of ethanol per day, respectively. Alcohol consumption was first converted to alcohol intake amount as g/day. We applied one unit as equal to 10 mL or 8 g of pure alcohol according to the National Health Service (NHS) instructions [30]. Details on the alcohol consumption calculations are described in Appendix A. BMI was divided into three groups: normal or underweight (BMI < 25), overweight (25 ≤ BMI < 30), and obese (BMI ≥ 30). The classification of BMI was also based on the NHS suggestion [31]. The WHR variable was initially calculated by dividing the waist circumference by the hip circumference and then categorizing into the normal group, which served as a reference (men: <0.90, women: <0.85), and the abdominal obesity group (men: ≥0.90, women: ≥0.85), according to the WHO recommendations [32]. The physical activity variable was measured as the summed metabolic equivalent of task (MET) minutes per week for all activity, provided by the UK Biobank. The MET categories (<600 MET-min/week, 600–3000 MET-min/week, and >3000 MET-min/week) were adapted from the 2018 WCRF/AICR Cancer Prevention Recommendations [28] and a previous UK Biobank study [33]. All medical history-related variables were obtained using records of self-reported medical conditions and ICD9 and ICD10 diagnoses in hospital records. All the exposure variables information is further described in Appendix A.

### 2.2. Derivation of the Polygenic Risk Score (PRS)

The polygenic score (PRS) in this study was constructed by using the single-nucleotide polymorphisms (SNPs) associated with PaCa from the GWAS Catalogue [18]. Only SNPs with reported effect size estimation and risk alleles from the GWAS Catalogue in any population were included. Thus, 96 SNPs were extracted from the UK Biobank imputation genotype file. This process was manipulated by QCTOOL [34] through the UK Biobank Research Analysis Platform (RAP), Swiss-Army-Knife APP. SNPs QC was conducted by PLINK1.9 [35] (excluding 24 SNPs, *n* = 72 SNPs). Only SNPs with a 90% genotyping rate (≤10% missing) were included. Next, we excluded SNPs that failed the Hardy–Weinberg test at a threshold of 10^−10^. Linkage disequilibrium (LD) was also assessed using the threshold for LD pruning (r^2^ > 0.8) (excluding 4 SNPs, *n* = 68 SNPs). This study did not include any ambiguous, palindromic SNPs or SNPs with an imputation score < 0.8 [36]. Finally, only the SNPs that passed the SNP QC from the Caucasian GWAS (*n* = 40 SNPs) were included in this study.

The sample QC was performed using STATA/MP software [37] version 17. Any individuals with missing SNPs information greater than 20% were excluded. Sex QC was evaluated by analyzing the difference between genetically inferred and self-reported sex. All the sex-chromosome aneuploidy were excluded. In the heterozygosity assessment, samples exhibiting outliers for heterozygosity (those falling outside ±3 standard deviations (SD) above or below the mean of the heterozygosity principal component analysis (PCA)-corrected values) were initially excluded. Subsequently, samples with poor heterozygosity or missingness, as suggested by the UK Biobank genomic analysis, were further removed. The identification of ancestral groups and assessment of relatedness to other participants in the dataset were conducted through PCA. Genetically Caucasian samples were verified based on their proximity to the mean, within ±3 SD of the first three genetic principal components. Ultimately, only genetically Caucasian samples were included in the established PRS and PaCa predictive model (*n* = 258,308).

The PaCa PRS was generated by the SNPs that passed the above SNP QC and were in the Caucasian GWAS (*n* = 40 SNPs). The individual weighted PRS values were calculated by summing the number of risk alleles of each SNP and then multiplying the sum by the effect size from the published GWAS. The weighted PRS values were applied to the classic PRS formula [38]:$$PRS=\beta_{1}\times {SNP}_{1}+\beta_{2}\times {SNP}_{2}+\cdots +\beta_{k}\times {SNP}_{k}\ldots +\beta_{n}\times {SNP}_{n}\;\;$$
where *β_k_* is the log odds ratio (OR) for *SNP_k_* from the previous GWAS, *SNP_k_* is the allele dosage for *SNP_k_*, and *n* is the number of *SNPs* included in this study. The standardized PRS was then derived by dividing each raw PRS by the SD of the raw PRS in the controls. Then, we employed quintile genetic risk classification, which was determined using the standardized PRS within the control group. The control standardized PRS served as the reference, with quintiles defined as follows: first quintile (Q1—up to 20%) PRS < 5.013, second quintile (Q2 > 20% to <40%) PRS ≥ 5.013 to <5.531, third quintile (Q3 > 40% to 60%) PRS ≥ 5.531 to <6.003, fourth quintile (Q4 > 60% to 80%) PRS ≥ 6.003 to <6.589, and fifth quintile (Q5 > 80%) PRS ≥ 6.589. Subsequently, PaCa cases were stratified into the PRS groups based on the quintile cut-off values.

The flowchart of SNPs extraction, SNPs QC, sample QC, and PRS construction is shown in Appendix A. The SNPs, risk alleles, and other summary statistics information used to extract SNPs from the UKB cohort and build the PRS are listed in Appendix A.

### 2.3. Statistical Analysis

The missing exposure variables were imputed by multivariate imputation using the chained equations (MICE) method [39]. A binary dependent variable was coded as PaCa cases and cancer-free controls. Independent exposure variables were classified as shown in Appendix A. In the demographic characteristic summary statistics, the Chi-square test was used for categorical data, and the Student’s *t*-test was used for continuous data to compare mean value differences between the cases and controls.

We applied a machine learning-based random forest method and a conventional multivariable logistic regression method to establish the PaCa prediction model for different purposes. Random forest is well known for dealing with high-dimensional and non-linear data and avoiding overfitting [40]. The random forest model was used for the purpose of identifying the most important risk factors. On the other hand, the multivariable logistic regression method was used to quantify the risks of the PaCa risk factors and interpret the PaCa probability in the prediction model.

In the random forest model, 85% of the dataset was initially set as a training dataset and 15% as a testing dataset. There was no clear guidance on determining the most optimal ratio for training and testing datasets [41]. However, the ideal ratio should depend on the unique characteristics and complexities of the dataset. Considering the properties of the large dataset in the UKB cohort and the relatively rare prevalent PaCa cases, we chose 85% of the dataset for training. In the consequential sensitivity test, we set 70% of the dataset as a training dataset and 30% as a testing dataset. The optimal hyperparameters were found in the training dataset via the RandomizedSearchCV [42] and GridSearchCV [43] functions in the Scikit-learn package by a 10-fold cross-validation process. The area under the receiver operating curve (AUC) was measured to determine the random forest model’s discriminatory power within the training and testing datasets. Internal validation was performed using a 10-fold cross-validation method to measure the confusion matrix (prediction, receiver operating characteristic (ROC) curve, and accuracy) between the training and testing datasets. Finally, model visualization of the feature importance was created to unveil the most influential variables. The feature importance was determined using the feature_importances_ function [44], which calculates the mean and standard deviation of the cumulative impurity decrease within each tree [44]. A summary SHapley Additive exPlanations (SHAP) plot [45] and a waterfall plot [46] were generated to illustrate how individual features contribute to the predictive outcome.

A multivariable logistic regression model was built by the forward and backward stepwise selection method to establish the risk prediction model. The multiple logistic regression model computed each risk variable’s OR. Due to the imbalance of case numbers to control numbers, we also performed a sensitivity test with three different ratios for the case and control groups, set at 1:10, 1:15, and 1:20. To visualize the multivariable logistic regression model, traditional and dynamic nomograms were created using the rms, nomogramEx, and DynNom R packages. The AUC was measured to evaluate the discrimination abilities of the multiple logistic regression model. The performance of the prediction model was confirmed by internal validation and calibration. Internal validation was performed via the 500 bootstrap resampling method, and the internal calibration was measured through the Hosmer–Lemeshow goodness-of-fit test to explore the agreement between the actual PaCa risk and the probability predicted risk.

A *p*-value less than 0.05 was considered “statistically significant,” and 95% confidence intervals (95% CI), not including one, were also used to guide the delineation of statistical significance. All statistical analyses were performed using STATA/MP software [37] version 17 (College Station, TX, USA: StataCorp LLC.), R software [47] (R version 4.2.1, R Development Core Team, Vienna, Austria), and Python [48] (version 3.10.12, Scotts Valley, CA: CreateSpace; 2009).

## 3. Results

After excluding the participants with other neoplasms, PaCa prevalent cases, and genetically non-Caucasian samples, a total of 258,308 participants (123,314 men and 134,994 women) were included in this study. The mean age of participants when entering the UKB cohort was 56.05 years (SD ± 8.10). Here, 960 participants were defined as PaCa incident cases (0.37%), and 257,348 participants were categorized as cancer-free controls (99.62%) (Table 1). There were missing data in four variables: blood type, WHR, tobacco smoking, and physical activity, before MISE imputation was manipulated. The missing data rates are shown in Appendix A.

### 3.1. Demographic Characteristic Distributions

The demographic characteristic distribution between the PaCa case group and the cancer-free control group is shown in Table 1. There was a slightly higher percentage of men (53.96%) as compared to women (46.04%) in the PaCa case group. The proportion of women (52.28%) was higher than men (47.72%) in controls. The mean age was higher in cases (~62 years of age) than in controls (~56 years of age) (Student’s *t*-test *p*-values < 0.05). In terms of PRS, the mean standardized PRS in cases was 6.12 (95% CI: 6.055–6.187), which was significantly higher than that in controls (5.84, 95% CI: 5.833–5.841) (Student’s *t*-test *p*-values < 0.05). Furthermore, the PRS quintile analysis revealed that more cases were distributed in the higher PRS quintile than in the lower PRS quintile. For blood type, there were more participants with a non-O blood type in cases than in controls (*p*-value < 0.05). There was no significant distribution difference between participants with a family history of bowel cancer in cases and controls.

For the lifestyle-related modifiable variables, the proportion of current smokers, previous smokers, men with alcohol consumption of more than 28 g per day, and women with alcohol consumption of more than 14 g per day were all higher in cases than in controls (*p*-value < 0.05). The proportion of participants with lower physical activity habits (<600 MET-minutes per week for all activity) was greater in cases compared to controls (*p*-value < 0.05). There was more participants who were overweight, obese, and abdominally obese in cases than in controls (*p*-value < 0.05).

Regarding the medical history-related variables, there was significantly more participants with a medical history of pancreatitis, DM, or gallbladder-related disease (cholecystitis, cholelithiasis, or cholecystectomy) in the PaCa cases than in the cancer-free controls (*p*-value < 0.05). Nonetheless, the distribution of hepatitis B, vitamin D deficiency, peritonitis, *H. pylori* infection, and SLE between cases and controls showed no significant differences. The distribution of demographic characteristics among the three different ratios for the case and control groups (Appendix A) was consistent with the results presented in Table 1.

### 3.2. Random Forest Results

A machine learning-based random forest model was also applied in this study. The optimal parameters were finally set as n_estimators = 1000, min_samples_split = 5, min_samples_leaf = 1, max_features = 4, and max_depth = 9, which were searched by RandomizedSearchCV [42] and GridSearchCV [43] through the 10-fold cross-validation method. To interpret the result of the random forest model, a visualized importance of features is depicted in Figure 1. The *x*-axis in Figure 1 indicates the importance of features, which corresponds to the mean decrease in impurity. The impurity-based feature importance method [44] was employed in this study for computing importance. In this method, the relative values of feature importance should be taken into consideration in the interpretation. As shown in Figure 1, the top ten influential variables were age, PRS, pancreatitis, DM, tobacco smoking, alcohol consumption, cholecystitis/cholelithiasis/cholecystectomy, BMI, physical activity, and gender. Among the top 10 risk variables, non-modifiable variables included age, PRS, and gender. Lifestyle-related modifiable risk variables were tobacco smoking, alcohol consumption, physical activity, and BMI. Medical history-related variables included pancreatitis, DM, and gallbladder disease (cholecystitis/cholelithiasis/cholecystectomy).

In the SHAP summary plot (Appendix A), higher age, higher PRS, the presence of DM, pancreatitis, gallbladder-related disease, current smoking, non-O blood type carriers, and consuming more alcohol contributed to an increased risk of PaCa. The summary SHAP plot further explained the feature effects on the feature importance plot (Figure 1). We also presented a case scenario in Appendix A, with a waterfall plot (Appendix A) as an example.

ROC curve analysis was performed to measure the discrimination power in the random forest model. The AUC was 0.88 and 0.77 in the training and testing datasets, respectively (Appendix A). Internal validation was evaluated using the 10-fold cross-validation method for calculating the mean values of prediction, ROC, and accuracy between the training and testing datasets (Appendix A). In Appendix A, prediction, ROC curve, and accuracy all showed similar values among the training and testing datasets.

In the sensitivity test, 70% of the dataset was set up as the training dataset and 30% as the testing dataset. Consequently, we obtained the same result for the top 10 risk variables (Appendix A). The discriminatory power and internal validation both performed well in this sensitivity test (Appendix A and Appendix A).

### 3.3. Pancreatic Cancer (PaCa) Risk Factors in the Multivariate Logistic Regression Model by Stepwise Selection

In the process of stepwise logistic regression analysis, a total of 18 variables were initially included and subsequently analyzed using both forward and backward stepwise selection methods. The same nine variables were seen in both approaches. The result of the multivariable logistic regression model is illustrated in Table 2. The established risk factors included male gender, age, non-O blood type, higher PRS, current smoker, higher alcohol consumption, medical history of pancreatitis, DM, and cholecystitis, cholelithiasis, or cholecystectomy.

The non-modifiable variables significantly associated with a higher PaCa risk included male gender (OR = 1.17, 95% CI: 1.02–1.33) relative to female gender, increasing per year of age (OR = 1.10, 95% CI: 1.07–1.51), non-O blood type carriers (OR = 1.29, 95% CI: 1.14–1.47) compared to O blood type carriers, and the higher PRS quintile participants. The odds of developing PaCa for participants in the fourth and fifth quintile PRS were 1.67 (95% CI: 1.35–2.07) and 2.03 (95% CI: 1.65–2.50), compared with the first quintile PRS.

Regarding the lifestyle-related modifiable variables, current tobacco smokers had a higher PaCa risk (OR = 1.82, 95% CI: 1.50–2.20) relative to never-smokers (Table 2). Compared to non-alcohol-drinking participants, the odds of PaCa were higher (OR = 1.27, 95% CI: 1.07–1.51) in the participants who consumed more alcohol (men > 28 g/d, women > 14 g/d).

In terms of the medical history-related variables, the odds of developing PaCa in the participants with a history of pancreatitis was 3.99 (95% CI: 3.06–5.22), relative to participants without a history of pancreatitis. Participants with DM history had a higher OR = 2.57 (95% CI: 2.21–2.99) than those without DM history. The odds of having PaCa among participants with a history of cholecystitis, cholelithiasis, or cholecystectomy were higher (OR = 2.07, 95% CI: 1.713–2.422) than for people without these histories. In the sensitivity test with various case and control ratios, the same nine variables were consistently selected within the three case-control ratio groups (case:control = 1:10, 1:15, and 1:20) during the stepwise logistic regression analysis. The results of the multivariable logistic regression model (Appendix A) also aligned with the findings presented in Table 2.

### 3.4. Model Performance

In the ROC curve analyses, the AUC of the PRS model was only 0.72 (95% CI: 0.708–0.737) (Appendix A). For the complete logistic model with the PRS variable, it was 0.78 (95% CI: 0.762–0.790), which is modestly higher than the model without the PRS variable (AUC = 0.76, 95% CI: 0.751–0.780) (Appendix A). The internal validation of the model performance via the 500-repetition bootstrap method is demonstrated in Appendix A. In terms of internal calibration, the Hosmer–Lemeshow goodness-of-fit test showed the predicted probability, and the actual internal estimates also fit well (*p*-value = 0.1543).

### 3.5. Traditional and Dynamic Nomograms

A visual nomogram prediction model based on the multivariable logistic model was constructed to predict the PaCa risk (Appendix A). The probability of PaCa can be obtained by summing the weighted point value of each risk variable on the scale (Appendix A). The nomogram demonstrated that a higher PaCa risk will be developed with a higher summarized total score. In addition, the nomogram revealed that increasing age contributed the most influence on the PaCa risk, followed by pancreatitis, DM, gallbladder-related disease (cholecystitis/cholelithiasis/cholecystectomy), PRS, tobacco smoking, alcohol consumption, blood type, and gender. The equation for calculating the total point value for each variable is presented in Appendix A.

Moreover, a dynamic nomogram was created for clinical or community applications, available at: https://ts35ky-temin-ke.shinyapps.io/DynNomapp/ (accessed on 6 October 2023). This website-based dynamic nomogram displays the probability of developing PaCa immediately on the right side of the screen after each parameter setting on the left side of the sketch map (Appendix A).

## 4. Discussion

To target the at-risk population of PaCa, our integrative PaCa risk prediction models were established by combining the non-modifiable variables, genetic predisposition variables, lifestyle-related modifiable variables, and medical history-related variables. Stepwise logistic regression and machine learning-based random forest models were applied to establish the prediction models. The most influential risk variables were revealed in visualizing the features’ importance (Figure 1). Dynamic nomograms (https://ts35ky-temin-ke.shinyapps.io/DynNomapp/ (accessed on 6 October 2023); Appendix A) were created for potential generalization in the clinical/community-based prevention program.

In our multivariable logistic regression models, the significant risk variables were as follows: non-modifiable—male gender and age, genetic predisposition—non-O blood type and a higher PRS quintile (Q4 vs. Q1, OR = 1.67; Q5 vs. Q1, OR = 2.03), lifestyle-related modifiable variables—current smokers and higher alcohol consumption, and medical history-related variables—pancreatitis, DM, and cholecystitis, cholelithiasis, or cholecystectomy. In previously published studies, a number of logistic regression models [20,21,49,50,51,52,53,54,55,56,57,58,59,60,61,62,63,64,65,66,67,68,69] were built to explore pancreatic cancer risks. Of them, nine models [20,53,56,57,62,63,64,66,68] were developed in the general population, and others were created in populations with DM history or clinical gastrointestinal symptoms. The common risk factors among the nine models [20,53,56,57,62,63,64,66,68] built in the general population were smoking, DM, pancreatitis, alcohol consumption, family history of PaCa, genetic predisposition, age, and sex. Nonetheless, no studies have evaluated a multi-domain model that integrates PRS, lifestyle-related modifiable variables, medical history-related variables, and other non-modifiable variables together in a single model such as ours. The study conducted by Salvatore et al. [68], using data from UKB and the Michigan Genomics Initiative (MGI), concentrated on the role of phenotype risk scores (PheRS) in the model with PRS, age, sex, and only a few lifestyle risk factors. Another risk model built by Klein et al. [57] contained genetic and non-genetic risk variables. However, their genetic variables only included three SNPs, and the only medical history considered was DM. Hence, our model sheds light on comprehensively measuring and integrating the multi-aspect risks of pancreatic cancer.

Regarding the model discrimination power, the ROC of our multivariable logistic regression model showed a value of 0.78, which was higher than previously reported studies [20,56,57,62,68]. In addition, most previous studies did not conduct model validation or calibration [20,53,56,57,62,63,64,66]. For their prediction model, Salvatore et al. [68] performed the Hosmer–Lemeshow test. Their result showed a significant difference in the calibration, whereas both the internal validation and calibration performed well in our prediction model.

Previous studies [19,20] have revealed that the PRS was associated with pancreatic ductal adenocarcinoma risk. In this study, higher PRS was presented as an essential risk of PaCa. Participants with the fifth quintile PRS have 2-fold odds for developing PaCa (*p* < 0.001) compared to those with the first quintile PRS. The AUC of the PRS alone model illustrated an excellent discrimination performance of 0.72 (Appendix A). On top of that, PRS also contributed to enhancing the PaCa model prediction power in this study. Compared to the logistic regression risk model without PRS (AUC = 0.76), the AUC of the model with PRS was slightly higher (AUC = 0.78) (Appendix A). One study, using PRS in the PaCa prediction from the UKB cohort [70], reported an AUC of 0.6 for their PRS model. However, some of the SNPs of their PRS were not from a Caucasian population GWAS, such as those from a Japanese study [62]. Since the majority of the population in the UKB cohort is Caucasian, the SNPs from other ethnicities would dilute the PRS prediction power. A diverse population may lead to different disease-related genetic variants [71]. Besides, the effect size value of each SNP was not from the published GWAS but instead from the UKB in the study by Sharma et al. [70]. Therefore, the effect size value still needs other GWAS studies to validate it. Another PRS model, based on the Pancreatic Disease Research Consortium, by Galeotti et al. [20] also presented the highest AUC of 0.6. In another UKB cohort study conducted by Kachuri [72], their SNPs were also only extracted from European-based GWAS, similar to our study. In accordance with our result, the AUC of PRS for PaCa prediction in their research was displayed as around 0.7. Hence, PRS is also useful for adding value to the PaCa prediction model, as well as the modifiable risk factors.

However, we also need to consider that the GWAS was produced by a large number of cases and controls. This may imply the power to detect the magnitude of risk differences between the groups of PRS. The role of PRS in the PaCa prediction has not been successfully demonstrated. One reason is that there was only 4.1% of explainable phenotypic variations from the current identified GWAS loci [73] for pancreatic cancer. As novel PaCa-associated variants continue to be investigated, the predictive value of PRS is anticipated to significantly improve. Furthermore, recent advancements, including methods such as combining PRS with high-penetrance genes [74,75], exploring gene–environment interactions, developing multifactorial scores [20], or incorporating phenotype risk scores (PheRS) [68], have the potential to enhance the predictive utility of PRS in clinical applications.

On the other hand, a machine learning-based random forest model was also used to predict the most crucial risk variables. In our random forest model, eight of the top ten influential variables were consistent with the logistic regression model stepwise selection result. The overlapped top five crucial features were echoed by the most influential features in the nomogram. The common risk variables between the two models are age, PRS, pancreatitis, DM, tobacco smoking, alcohol consumption, cholecystitis/cholelithiasis/cholecystectomy, and gender, the top four of which were age, PRS, pancreatitis, and DM.

Regarding the discrimination power of the random forest model, the AUC performance was good in the training and testing datasets (Appendix A), respectively. The 10-fold cross-validation also showed high prediction and accuracy in the model performance (Appendix A). To our knowledge, there are no other random forest PaCa prediction models in the previous studies. A study based on an artificial neural network (ANN) model was developed by Muhammad et al. [76]. However, it is difficult to identify the most critical features in that ANN model. Therefore, compared to our random forest model, it is less accessible for an ANN model to be applied to raise population awareness and stratify the high-risk population by showing the most significant risk factors.

Other machine learning models, including support vector machine (SVM) and gradient boosting (GB), were also considered for application in clinical prediction models [77]. SVM is one of the supervised learning methods commonly used for addressing classification problems, particularly effective with unstructured and semi-structured data [78]. However, considering the specific requirements of this study, which involve the quantification of risk factors for interpretation, we opted for the logistic regression model. In comparison to random forest (RF), GB trains trees sequentially by correcting errors from previous trees [79]. In addition, GB offers more parameters for fine-tuning [79]. Although GB can potentially achieve optimal performance compared to RF, it tends to be more sensitive to overfitting [80]. Notably, one of the main advantages of RF is its resistance to overfitting [40]. Hence, for this study, we employed the random forest model. Although it is recommended to explore a novel ensemble learning model that combines different clinical machine learning models and image-based deep learning models to forecast postoperative survival in pancreatic cancer patients [81], in our study, we did not have image data available to explore this novel approach.

Furthermore, the nomogram (Appendix A) displayed the PaCa probability risk by summing each risk variable point value. The dynamic nomogram (Appendix A) could make the PaCa risk prediction model more accessible, feasible, and convenient in clinical or community-based prevention programs. To our knowledge, no other PaCa risk prediction models have been converted to nomograms previously.

Comprehensively, there are different advantages and disadvantages of our two models. The random forest model possesses the benefit of reducing overfitting and handling the many variables, which can provide us with the most impactful risk factors. However, when considering increasing the availability, feasibility, and accessibility of the PaCa risk prediction, we suggest using the multiple logistic regression model to obtain the OR information for the significant risk variables. This can then be applied to develop a dynamic nomogram for generalized use in clinical, primary care, or community prevention practice programs.

A systematic review [21] of pancreatic cancer models shows that a large number of models have been previously built. Nevertheless, there is still a lack of a comprehensive model integrating lifestyle-related modifiable risk variables, genetic-related variables, and medical history-related variables for clinical and community application. Our integrated PaCa prediction model can not only identify the most influential risk variables of PaCa and show the odds ratio of each risk variable but can also be used practically to demonstrate the risk of developing PaCa in the population. This integrated PaCa prediction model will also be applied in our community-based iHelp project study.

## 5. Strengths and Limitations

The strengths of this study were, first, the UKB cohort has a large population sample of half a million across the UK. The exposure data contained comprehensive, in-depth genetic and health information. All the screening questionnaires, genetic data, and disease status information are robust and of a high quality. Second, our PRS building was strictly constrained to the SNPs that passed the SNPs QC and samples that passed the sample QC. Moreover, only the SNPs from the Caucasian GWAS were considered in this study. Thus, our PRS was built under a high-quality selection process. Third, to our knowledge, this is the first PaCa prediction model integrating genetic-related, lifestyle-related, and medical history-related variables with robust discrimination. Ultimately, the result was visualized in a dynamic nomogram, which provides ease-of-use for applications in clinical practice.

There are still some limitations in our study. First is the healthy volunteer selection bias in the UK Biobank, which has been considered previously [82,83]. Therefore, the risk estimates of some of the variables may be underestimated compared to the general population in the UK. This applies particularly to the lifestyle-related variables, such as smoking and alcohol drinking. However, the genetic-related risk may be less influenced by the healthy volunteer selection bias. In addition, the limitation of the currently explored variables may restrict our PRS prediction power. To upgrade our PaCa predictive value, we would consider combining the highly penetrating variables in the future. Another limitation is that pancreatic cancer is relatively rare; therefore, the number of cases within the follow-up timeframe was fewer than 5% of the whole cohort. This could have an impact on the estimated probability [84]. Next, external validation of our predictive model was constrained by the challenge of obtaining an appropriate large dataset that included an adequate number of pancreatic cancer cases and all the variables presented in our model. However, internal validity, assessed through bootstrapping, as an indicator of wider validity [85], can also be considered as an approximation to external validity [86]. Finally, the PRS used in this study was constructed exclusively from SNPs identified in the Caucasian GWAS. Moreover, the study samples were exclusively drawn from the Caucasian population. It is crucial to exercise caution when extrapolating our findings to non-Caucasian populations, as the genetic basis for PaCa may vary across different ethnic groups.

## 6. Conclusions

Our integrative PaCa risk prediction model, developed using data from the UK Biobank, allows for effectively stratifying individuals at future risk of PaCA in the UK population. The inclusion of a polygenic score enhanced the discriminative power of our PaCa risk prediction model. While pancreatic cancer remains a lethal disease without established prevention or screening programs, our model enables the identification of at-risk individuals within the community, thereby facilitating targeted prevention efforts. Our findings, encompassing non-modifiable genetic and non-genetic factors, lifestyle-related variables, and medical history-related factors, can raise awareness of pancreatic cancer risk factors among the public. Additionally, these risk variables can serve as a reference for further etiological investigations. Moreover, incorporating the dynamic nomogram into our visualization model will further support our ongoing community-based cancer prevention initiatives, aimed at motivating community members to seek early prevention for pancreatic cancer. 

## Figures and Tables

**Figure 1 biomedicines-11-03206-f001:**
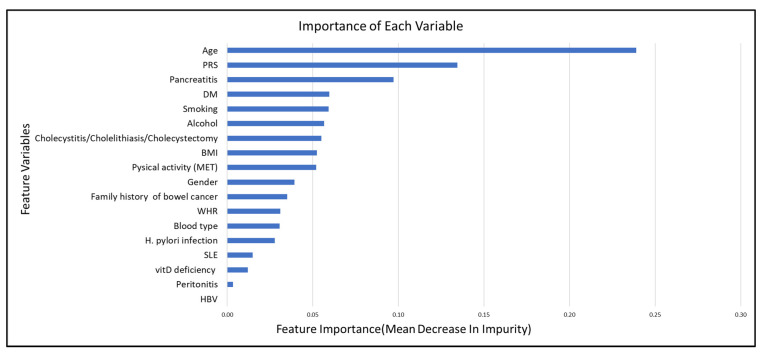
The importance of each variable.

**Table 1 biomedicines-11-03206-t001:** Demographic characteristics of pancreatic cancer (PaCa) cases and cancer-free controls.

Characteristic Variables	Pancreatic Cancer (PaCa) Cases (*n =* 960)	Cancer-Free Controls (*n* = 257,348)	*p*-Value *
**Gender**			<0.001
Woman	442 (46.04%)	134,552 (52.28%)	
Man	518 (53.96%)	122,796 (47.72%)	
**Age ^#^**	61.60	56.03	<0.001
**Polygenic score (PRS) ^#^(con)**			
Standardized PRS	6.12	5.84	<0.001
**Polygenic score (PRS) (cat)**			<0.001
Q1	135 (14.06%)	51,469 (20%)	
Q2	141 (14.69%)	51,470 (20%)	
Q3	168 (17.50%)	51,469 (20%)	
Q4	234 (24.38%)	51,470 (20%)	
Q5	282 (29.38%)	51,470 (20%)	
**Blood type**			<0.001
O blood type	355 (36.98%)	112,083 (43.55%)	
Non-O blood type	605 (63.02%)	145,265 (56.45%)	
**Family history of bowel cancer**			0.255
No	858 (89.38%)	232,791 (90.46%)	
Yes	102 (10.63%)	24,557 (9.54%)	
**Tobacco smoking status**			<0.001
Never	441 (45.94%)	143,911 (55.92%)	
Previous	373 (38.85%)	87,948 (34.17%)	
Current	146 (15.21%)	25,489 (9.90%)	
**Alcohol consumption amount**			0.192
Never	226 (23.54%)	57,662 (22.41%)	
Men: >0–≤28 g/d, Women: >0–≤14 g/d	347 (36.15%)	100,383 (39.01%)	
Men: >28 g/d, Women: >14 g/d	387 (40.31%)	99,303 (38.59%)	
**Physical activity (MET-min/week)** ^a^			0.012
<600	206 (21.46%)	47,721 (18.54%)	
600–3000	444 (46.25%)	130,485 (50.70%)	
>3000	310 (32.29%)	79,142 (30.75%)	
**BMI**			<0.001
Normal or underweight (BMI < 25)	249 (25.94%)	85,040 (33.04%)	
Overweight (25 ≤ BMI < 30)	421 (43.85%)	109,575 (42.58%)	
Obese (BMI ≥ 30)	290 (30.21%)	62,733 (24.38%)	
**Waist–hip ratio (WHR)**			<0.001
Normal (Men: <0.90, Women: <0.85)	372 (38.75%)	131,445 (51.08%)	
Abdominal obesity (Men: ≥0.90, Women: ≥0.85)	588 (61.25%)	125,903 (48.92%)	
**Medical history-related variables**			
**Pancreatitis**			<0.001
No	888 (92.05%)	254,962 (99.07%)	
Yes	72 (7.5%)	2386 (0.93%)	
**Diabetes mellitus**			<0.001
No	713 (74.27%)	235,818 (91.63%)	
Yes	247 (25.73%)	21,530 (8.37%)	
**Hepatitis B**			0.374
No	960 (100%)	257,136 (99.92%)	
Yes	0 (0%)	212 (0.08%)	
**Cholecystitis/cholelithiasis/cholecystectomy**			<0.001
No	761 (79.27%)	237,575 (92.32%)	
Yes	199 (20.73%)	19,773 (7.68%)	
***Helicobacter pylori* infection**			0.229
No	949 (98.85%)	255,291 (99.20%)	
Yes	11 (1.15%)	2057 (0.80%)	
**Systemic lupus erythematosus (SLE)**			0.607
No	959 (99.90%)	256,902 (99.83%)	
Yes	1 (0.10%)	446 (0.17%)	
**Vitamin D deficiency**			
No	951 (99.06%)	255,368 (99.23%)	0.552
Yes	9 (0.94%)	1980 (0.77%)	
**Peritonitis**			0.157
No	959 (99.90%)	256,349 (99.61%)	
Yes	1 (0.10%)	999 (0.39%)	

^#^ Mean value; * Chi-square test statistic or Student’s *t*-test for mean values; ^a^ 600 MET-min/week = 150 min/week; 3000 MET-min/week = 750 min/week.

**Table 2 biomedicines-11-03206-t002:** Pancreatic cancer (PaCa) risk factors in the multivariable logistic regression model.

Characteristic Variables	OR	95% CI	*p*-Value
**Non-modifiable variables**			
** Gender**			
Woman	Ref.		
Man	1.17	(1.02–1.33)	0.024
** Age**	1.10	(1.07–1.51)	<0.001
**Blood type**			
O blood type	Ref.		
Non-O blood type	1.29	(1.14–1.47)	<0.001
**Polygenic score (PRS)**			
Q1	Ref.		
Q2	1.05	(0.83–1.33)	0.690
Q3	1.22	(0.97–1.53)	0.091
Q4	1.67	(1.35–2.07)	<0.001
Q5	2.03	(1.65–2.50)	<0.001
**Lifestyle-related modifiable variables**			
**Tobacco smoking status**			
Never	Ref.		
Previous	1.01	(0.88–1.16)	0.906
Current	1.82	(1.50–2.20)	<0.001
**Alcohol consumption amount**			
Never	Ref.		
Men > 0–28 g/d, Women > 0–14 g/d	1.01	(0.85–1.21)	0.873
Men > 28 g/d, Women > 14 g/d	1.27	(1.07–1.51)	0.005
**Medical history-related variables**			
**Pancreatitis**			
No	Ref.		
Yes	3.99	(3.06–5.22)	<0.001
**Diabetes mellitus (DM)**			
No	Ref.		
Yes	2.57	(2.21–2.99)	<0.001
**Cholecystitis/cholelithiasis/cholecystectomy**			
No	Ref.		
Yes	2.04	(1.71–2.42)	<0.001

## Data Availability

Data can only be accessed from the UK Biobank via the approved application. The data is owned by the UK Biobank (www.ukbiobank.ac.uk (accessed on 5 December 2022)) and as researchers we are not entitled to republish or otherwise make available any UK Biobank data at the individual participant level. The UK Biobank, however, is open to all bona fide researchers anywhere in the world. Detailed access procedures can be found by following this link: https://www.ukbiobank.ac.uk/enable-your-research/apply-for-access (accessed on 26 October 2023). The data used in this study (application number 94611) can be requested by applying through the UK Biobank Access Management System (www.ukbiobank.ac.uk/register-apply (accessed on 26 October 2023)).

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
