# Peer review of "An Integrative Pancreatic Cancer Risk Prediction Model in the UK Biobank"

_biomedicines, 2023, doi:10.3390/biomedicines11123206_

Round 1

Reviewer 1 Report

Comments and Suggestions for Authors

The authors, along with their collaborators, developed a model for predicting pancreatic cancer utilizing an extensive dataset from the genetic analyses of UKB, and artificial intelligence. The model showcases commendable performance, marking a noteworthy accomplishment. The paper holds significance in illustrating the potential of genome analysis as a pivotal tool in cancer prediction. However, certain aspects merit attention.

1)     Pancreatic cancer exhibits a notably low prevalence, and conventional models, even those integrating artificial intelligence, have demonstrated suboptimal performance. The paper achieves high performance, and it is imperative to elucidate the underlying reasons and ascertain whether additional statistical processing contributed to this success. Typically, the adjustment of patient and control group sizes is common in light of performance constraints; hence, an exploration of this aspect is warranted.

2)     The paper highlights the utilization of SNPs from the GWAS catalogue and the exclusive selection of patients with European ancestry due to outliers of population structure. The discussion should address the potential implications of excluding studies, which reported SNPs in GWAS catalogue, focused on African or Asian populations. Given the European-centric focus of the study, it is crucial to explicitly acknowledge its limited applicability.

3)      Upon examination of Table 1, a constant 20% PRS score is observed in cancer-free patients. Verification of these numbers is essential. If accurate, an elucidation on whether 20% of the general population possesses bunch of variants is required. Additionally, the discussion notes a consistent high risk of pancreatic cancer in the general population, warranting clarification, especially in relation to the observed odds ratio of 1.67 (Q4 vs. Q1) in your discussion. Or correct number in table 1.

4)     The assertion of the random forest model's superiority warrants consideration, especially in light of the prevailing practice in other studies that often employ multiple models and ensemble methods due to the inherent limitations of individual models. A discussion on the potential merits of employing alternative models is recommended. (References (PMID): 36049618, 3094880).

Minor Point:

1)     A reevaluation of the percentage of family history of bowel cancer is necessary.

2)     Concerning Figure 1, clarification on the meaning of the X-axis is required. The addition of a detailed description is crucial, as is customary in the presentation of SHAP summary plots, which typically include negative effects of variables

3)     Regarding PRS, clarification on the exclusion criterion related to target quality control is needed. Specifically, if the authors intended to convey heterozygosity greater than 3SD, this should be explicitly stated.

Reviewer 2 Report

Comments and Suggestions for Authors

The abstract needs quantification. The introduction needs tree diagram. Attributes relationship are to shown by some statistical tests like Fried man test. Correlation analysis may be included. Limitation of the paper may be included.  The Random forest may be compare with SVM. How the class imbalance were solved. 

Round 2

Reviewer 2 Report

Comments and Suggestions for Authors

All the corrections are included in the paper. There is no need for further review.

Comments on the Quality of English Language

Nil